# ARMTRAJ: A Set of Multi-Purpose Trajectory Datasets Augmenting the Atmospheric Radiation Measurement (ARM) User Facility Measurements

Israel Silber[1], Jennifer M. Comstock[2], Michael R. Kieburtz[2], Lynn M. Russell[3]

[1] Atmospheric, Climate, and Earth Sciences Division, Pacific Northwest National Laboratory, Richland, WA, 99352, USA
[2] Advanced Computing, Mathematics, and Data Division, Pacific Northwest National Laboratory, Richland, WA, 99352, USA
[3] Scripps Institution of Oceanography, University of California, San Diego, CA, 92093, USA

*Correspondence to*: Israel Silber (israel.silber@pnnl.gov)

**Abstract.** Ground-based instruments offer unique capabilities such as detailed atmospheric thermodynamic, cloud, and aerosol profiling at a high temporal sampling rate. The U.S. Department of Energy Atmospheric Radiation Measurement (ARM) user facility provides comprehensive datasets from key locations around the globe, facilitating long-term characterization and process-level understanding of clouds, aerosol, and aerosol-cloud interactions. However, as with other ground-based datasets, the fixed (Eulerian) nature of these measurements often introduces a knowledge gap in relating those observations with airmass hysteresis. Here, we describe ARMTRAJ, a set of multi-purpose trajectory datasets that helps close this gap in ARM deployments. Each dataset targets a different aspect of atmospheric research, including the analysis of surface, planetary boundary layer, distinct liquid-bearing cloud layers, and (primary) cloud decks. Trajectories are calculated using the Hybrid Single-Particle Lagrangian Integrated Trajectory (HYSPLIT) model informed by the European Centre for Medium-Range Weather Forecasts ERA5 reanalysis dataset at its highest spatial resolution (0.25 degrees) and are initialized using ARM datasets. The trajectory datasets include information about airmass coordinates and state variables extracted from ERA5 before and after the ARM site overpass. Ensemble runs generated for each model initialization enhance trajectory consistency, while ensemble variability serves as a valuable uncertainty metric for those reported airmass coordinates and state variables. Following the description of dataset processing and structure, we demonstrate applications of ARMTRAJ to a case study and a few bulk analyses of observations collected during ARM's Eastern Pacific Cloud Aerosol Precipitation Experiment (EPCAPE) field deployment. ARMTRAJ will soon become a near real-time product accompanying new ARM deployments and an augmenting product to ongoing and previous deployments, promoting reaching science goals of research relying on ARM observations.

## 1 Introduction

Synergistic use of ground-based, airborne, and satellite observations with continuously improving models promotes a better understanding of cloud and aerosol source and sink processes, aerosol-cloud interactions (ACI), and cloud-climate feedbacks, and helps refine climate projection. Nevertheless, even though high-resolution and Earth system models are becoming more sophisticated, our knowledge about some of these multi-scale processes and their associate intensities and rates is still deficient; ergo, they remain the leading source of uncertainty in climate model predictions (Forster et al., 2021). Cutting-edge ground-based observations and their synthesis thereof provide opportunities to study cloud and aerosol processes in great detail. The US Department of Energy Atmospheric Radiation Measurement (ARM) user facility operates multiple comprehensive suites of such instruments, which are deployed to key locations around the globe, including, for example, densely-populated urban environments and high-interest regions such as the Southern Ocean, the Antarctic and Arctic, and the Atlantic and Pacific oceans' upwelling regions (Dorsey et al., 2024). ARM's mobile, fixed, and aerial facilities include, among other instruments, surface aerosol observing systems covering a wide range of sizes and properties (Uin and Smith, 2020), a range of profiling and scanning radars and lidars (e.g., Muradyan and Coulter, 2020; Widener et al., 2012a, b; Widener and Bharadwaj, 2012), and uncrewed aerial vehicles (Schmid and Ivey, 2016), which promote atmospheric dynamic, thermodynamic, and radiative process research with a specific focus on clouds and aerosols.

In recent years, there has been a growing community recognition of the importance of Lagrangian considerations for acquiring causal understanding wherever atmospheric dynamics play a crucial role. This recognition is manifested in the integration of Lagrangian components in numerous aerosol and cloud, observational, and/or model simulation-based studies. For example, a comprehensive understanding of cloud lifecycles often necessitates knowledge about the hysteresis and origin of cloudy airmasses. Trajectory analyses support studies focused on warm, mixed-phase, and cold clouds, from low to high latitudes (e.g., Christensen et al., 2020; Ilotoviz et al., 2021; Mohrmann et al., 2019; Silber and Shupe, 2022; Svensson et al., 2023; Wernli et al., 2016). Back-trajectories can inform on potential cloud formation mechanisms (e.g., Silber and Shupe, 2022; Svensson et al., 2023), be used to evaluate the influence of airmass intrusions on cloud evolution (e.g., Christensen et al., 2020; Ilotoviz et al., 2021; Mohrmann et al., 2019), and generally support process understanding through modeling studies by providing boundary conditions and observationally-based benchmarks (e.g., Neggers et al., 2019; Silber et al., 2019; Tornow et al., 2022).

Estimations of airmass trajectories and origin are also highly valuable for understanding aerosol hysteresis and potential indirect effects. For example, back-trajectory analyses were previously used to examine dust ice nucleating particle (INP) processing before entraining into a cloud (e.g., Wiacek and Peter, 2009), to quantify periods of chemical reactions experienced by aerosol prior to their transport to ground-based stations (e.g., Hawkins and Russell, 2010), to estimate source contribution functions and similarity of source regions (e.g., Day et al., 2010; Liu et al., 2011), and to study long-range transport of aerosols onto surface sites (e.g., Zheng et al., 2020).

While ARM field deployments provide a high-end, unique, and comprehensive suite of measurements, most samples are collected at fixed sites; that is, from an Eulerian perspective. Certain field campaigns include multiple deployment sites along climatological flow patterns (e.g., Geerts et al., 2022), yet a knowledge gap often still exists, which can be ameliorated using trajectory calculations. Here we describe ARMTRAJ, a set of Lagrangian trajectory data products for ARM fixed sites and mobile deployments, which can be used to close some of the gaps ensuing from the Eulerian nature of many ARM cloud, aerosol, and other atmospheric measurements, thereby enhancing the versatility of ARM datasets. For example, understanding the impact of pollution upwind of ARM deployment sites on measured aerosol properties versus clean upwind conditions; evaluating the effect of cloud-top aerosol entrainment on sink processes by using ARMTRAJ data and ARM measurements to initialize and force model simulations; and analyzing cloud lifecycles before and after overpassing ARM sites by synthesizing ARM, satellite, and ARMTRAJ data. ARMTRAJ is based on the HYSPLIT model (Stein et al., 2015) informed by the European Centre for Medium-Range Weather Forecasts (ECMWF) ERA5 reanalysis (Hersbach et al., 2020) at its highest spatial resolution (0.25 degrees; ~31 km). ARMTRAJ datasets provide information about airmass coordinates upwind and (in certain datasets) downwind, together with their thermodynamic properties and overpassed surface characteristics. Varying-size ensemble run results are also reported, facilitating the evaluation of trajectory consistency, robustness, and uncertainty while mitigating potential near-surface artifacts and errors. In sect. 2, we describe ARMTRAJ's four dataset types, focusing on the surface, planetary boundary layer (PBL), and observed clouds over ARM sites. In sect. 3, we demonstrate ARMTRAJ dataset applications using ARMTRAJ data for the ARM Eastern Pacific Cloud Aerosol Precipitation Experiment (EPCAPE; Russell et al., 2021) available on the ARM Data Discovery (https://adc.arm.gov/discovery/). Conclusions and a short outlook are given in sect. 4.

## 2 ARMTRAJ Processing and Dataset Structure

ARMTRAJ's four datasets include a surface, PBL, liquid cloud layer, and primary cloud deck datasets, hereafter referred to as ARMTRAJ-SFC, ARMTRAJ-PBL, ARMTRAJ-CLD, and ARMTRAJ-ARSCL. Datasets are organized in daily files. Each file is in NetCDF format and follows ARM standards (see Palanisamy, 2016), including full metadata for each variable field. Each dataset contains sets of the following variables and properties extracted and derived from ERA5 data along each airmass trajectory:

- Date and time.
- Airmass coordinates: latitude, longitude, altitude above mean sea level (AMSL), and height above ground level (AGL).
- Thermodynamic variables: airmass pressure, temperature, potential temperature, equivalent potential temperature (excluding condensate from the calculation), virtual potential temperature, specific humidity, relative humidity (RH), and RH with respect to ice.
- Hourly-mean airmass ascent rate (vertical motion).

- PBL height (PBLH) in airmass column and airmass height-to-PBLH ratio (greater than 1 when airmass is above the PBL, and vice versa).
- Land-sea mask (land area fraction) and daily (00 UTC) sea-ice cover in airmass column (based on ERA5's associated ~31 km native grid-cell).
95
- Other surface properties in the airmass column: terrain orientation and distortion in the horizontal plane, standard deviation and slope of orography within the ERA5 grid cell (using a minimum horizontal feature scale of 5 km), low and high vegetation type and cover, and soil type.

Trajectory calculations are performed with HYSPLIT, reading the same ERA5 global data files in pressure-level vertical coordinates supplemented with single-level reanalysis data fields such as PBLH and surface altitude, winds, and roughness
100 length. Each ARMTRAJ dataset is initialized and configured differently to align with its purpose, potential use, and the characteristics of the ARM dataset required for initialization (see Table 1). ARMTRAJ datasets are discussed in detail below.

**Table 1: ARMTRAJ dataset summary. The dataset names ARMTRAJ-SFC, ARMTRAJ-PBL, ARMTRAJ-CLD, and ARMTRAJ-ARSCL refer to the surface, planetary boundary layer, liquid cloud layer, and primary cloud deck datasets, respectively.**

| Dataset name | Initialized at | Initialization time | Includes a free tropospheric run | Ensemble size | Back trajectory period | Forward trajectory period | Potential application examples |
|---|---|---|---|---|---|---|---|
| ARMTRAJ -SFC | surface | 3-h increments (00, 03, 06, … UTC) | No | 18 | 10 days | - | Long-range aerosol transport; estimation of periods of chemical reactions |
| ARMTRAJ -PBL | 11 equally distant heights from the surface to the PBLH | same as ARM radiosondes | Yes | 99[*] | 5 days | - | PBL airmass hysteresis (aerosol sources, interactions with the surface, etc.) |
| ARMTRAJ -CLD | center of each detected cloud layer | same as ARM radiosondes | No | 27[**] | 5 days | 5 days | Evaluation of cloud formation mechanisms; boundary conditions for model simulations |
| ARMTRAJ -ARSCL | 11 equally distant heights between the hourly mean base and top of the lowest (typically primary) cloud deck | 3-h increments (00, 03, 06, … UTC) | Yes | 99[*] | 5 days | 5 days | Cloud deck and free tropospheric (entrained) airmass sources; boundary conditions for model simulations |

[*] Ensemble size of 9 in free-tropospheric runs (see sect. 2.2)

[**] Per detected liquid-bearing cloud layer (see sect. 2.3)

105

## 2.1 Surface Trajectory Dataset (ARMTRAJ-SFC)

The ARMTRAJ-SFC dataset is designed to support research using ARM's surface measurements, with an emphasis on aerosol observations. While ground-based remote-sensing measurements and retrievals are made regularly and airborne in-situ aerosol observations occur episodically during intensive observing periods, surface measurements are typically the most informative about the sampled aerosol chemical, morphological, microphysical, and radiative properties given fewer limitations such as payload dimensions and weight. For a given day at a given ARM site, ARMTRAJ-SFC is initialized at the surface every 3 hours. Each run includes a 10-day back trajectory. The 10-day period is sufficient to determine potential long-range aerosol transport sources and/or estimate periods of relevant chemical reactions (e.g., Hawkins and Russell, 2010; Lata et al., 2021; Zheng et al., 2020). While some studies examined back trajectories extending even 15 days, tests we performed using an ensemble approach (not shown) suggested that trajectory dispersion predominantly becomes so substantial that the airmass information is no longer consistent nor robust. This dispersion is most likely driven by the propagation of errors stemming from multiple factors, such as the integration time step and the limited vertical resolution of the ERA5 pressure-level data used by HYSPLIT, especially near the surface.

In addition to the information for the trajectory initialized at the ARM site (surface level), the mean and standard deviation of ensemble results are reported for each of the variables listed above, except for the orographic, vegetation, and soil properties, the values of which are reported for the ensemble mean coordinates. The ensemble is initialized using two starting heights (surface and 50 m AGL) and 9 starting horizontal locations (combinations of site coordinates ± 7.5 km in the east-west and north-south directions, defining a 3×3 grid) for a total of 18 ensemble members. The fixed geodetic distance in metric units rather than in arc degrees is used to ensure ARMTRAJ's ensemble configuration consistency when initialized in different geographic regions; for example, in ARM's North Slope of Alaska site (Verlinde et al., 2016), where a given longitudinal arc length translates to shorter geodetic distances relative to lower-latitude sites. The ensemble starting horizontal extent covers roughly half of the horizontal dimension of ERA5 grid cells, allowing the evaluation of ensembles' physical variability yet keeping them initially constrained to the site vicinity. In practice, the ensemble results, specifically the standard deviation of reported ensemble variables, can be treated as a measure of trajectory estimated uncertainty and potentially serve as tests for general trajectory robustness. We note that ARMTRAJ-SFC data files are supplemented with 1-hour mean and standard deviation values (starting at trajectory initialization time) of surface observations from the corresponding ARM site Surface Meteorology System (Ritsche, 2011).

## 2.2 Planetary Boundary Layer Trajectory Dataset (ARMTRAJ-PBL)

ARMTRAJ-PBL, which could support PBL cloud and aerosol research in addition to other PBL research topics, includes 5-day back trajectory calculations for the base (surface), middle, and top of the PBL (i.e., the PBLH). The PBLH used in HYSPLIT initialization is determined from ARM radiosonde measurements (Holdridge, 2020) using a bulk Richardson number method (Troen and Mahrt, 1986; Vogelezang and Holtslag, 1996) with a critical threshold value of 0.25, as reported

in ARM's PBLH value-added product (VAP) (Sivaraman et al., 2013). Therefore, ARMTRAJ-PBL trajectories are initialized at radiosonde release times rounded to the nearest whole hour, resulting in 2 to 4 trajectory starting times for a given day, depending on sounding measurement availability.

There are other methods to determine the PBLH, the radiosonde-based retrievals of which are reported in ARM's VAP (see Sivaraman et al., 2013) and included in ARMTRAJ-PBL. However, the utilized bulk Richardson number method and its threshold value were evaluated by Seidel et al. (2012), who suggested they are suitable for both convective and stable PBLs, though we note that Zhang et al. (2022) recently suggested this method better compares to ceilometer-based PBLH determination method under stable PBL conditions. Moreover, the same method and threshold values are consistent with the PBLH implementation in ERA5 diagnostics used here.

The ensemble in the ARMTRAJ-PBL dataset is much greater than ARMTRAJ-SFC's ensemble. It consists of 11 equally distant heights from the surface to the PBLH combined with a similar 3×3 grid, totaling 99 ensemble members. This extensive ensemble configuration ameliorates the lack of explicit mixing in the ECMWF Integrated Forecasting System (IFS) model used to drive ERA5 and the limited near-surface resolution (~250 m) at the ERA5 pressure level grid.

The PBL and its associated cloud and aerosol fields are known to interact with the free troposphere above the PBLH, with a potentially pronounced indirect impact on properties and processes such as cloud lifecycles, aerosol scavenging, and the PBLH (e.g., Jiang et al., 2002; Raes, 1995; Raes et al., 2000; Sorooshian et al., 2020; Tornow et al., 2022). ARMTRAJ-PBL also includes free-tropospheric runs for each trajectory starting time, initialized 200 m above the reported PBLH to support and augment studies focusing on free-tropospheric entrainment effects. The free-tropospheric run results include all variables listed at the beginning of this section, as well as results for a small 9-member ensemble, initialized at the same height with the 3×3 grid as in the other ensembles.

### 2.3 Liquid-Bearing Cloud Layer Trajectory Dataset (ARMTRAJ-CLD)

ARMTRAJ-CLD aims to augment liquid-bearing cloud studies from warm to mixed-phase clouds. The 5-day back and forward trajectories reported in this dataset broadly cover the typical residence time of moisture in the atmosphere (see Ent and Tuinenburg, 2017; Läderach and Sodemann, 2016; Woods and Caballero, 2016) and can, therefore, promote the evaluation of cloud formation mechanisms and cloudy airmass hysteresis. The dataset provides essential information for the configuration and initialization of modeling exercises (e.g., Silber et al., 2019; Tornow et al., 2022), and in some cases, the reported trajectories can even inform on other overpassed ground-based observational sites upwind or downwind (e.g., Ali and Pithan, 2020), further constraining modeling efforts.

Like ARMTRAJ-PBL, ARMTRAJ-CLD's initialization depends on ARM radiosonde data (see Table 1) for determining liquid-bearing cloud layers and, therefore, has the same starting times. A leading advantage of using radiosonde-based cloud detections for initialization is that we can examine full tropospheric profiles and are not confined to the first few to several optical depths, as in the case of lidars that are commonly used to detect liquid-bearing cloud layers from the bottom-up in the

case of ground-based observations (or top-down in the case of satellite and aircraft observations). Liquid-bearing cloud layers
       are determined from radiosonde RH profiles using the following steps:

1.  Set radiosonde samples as "cloud" if RH values exceed 96%. This threshold value considers the radiosonde vendor's
    uncertainty (Holdridge, 2020) and was previously validated using lidar-based cloud layer detections (Silber et al.,
    2020, Fig. S1; e.g., Silber and Shupe, 2022; Stanford et al., 2023, Appendix D). We have also qualitatively examined
detection consistency using higher RH thresholds against other remote-sensing measurements for different observed
    cases (not shown) and came to the same conclusion regarding the 96% threshold value validity.
2.  Concatenate "cloud" samples vertically distant by less than 50 m from each other.
3.  Remove resulting layers if their total thickness (including the thickness of the cloud-top sample) is smaller than 25
    m.

ARMTRAJ-CLD can report trajectories for up to 10 detected overlying liquid-bearing layers per initialization time step. The
       initialization height is set to the center of each detected cloud layer. In this case, the ensemble results are based on 27 members
       per detected cloud layer: 3 vertical starting heights (cloud layer center and center ± 50 m), and 9 horizontal coordinates using
       the same 3×3 grid as in the other datasets. The detected liquid-bearing cloud layer boundaries and the utilized radiosonde
       thermodynamic and wind measurements are also reported in ARMTRAJ-CLD to support cloud-related analysis further.

**2.4 Primary Cloud Deck Trajectory Dataset (ARMTRAJ-ARSCL)**

       Many studies such as those on marine stratocumulus clouds, often focus on primary cloud decks, which in this context refer
       to the optically and geometrically thickest cloud decks in atmospheric columns. Those primary cloud decks typically produce
       a significant radiative effect, impacting the surface and atmospheric energy budgets. ARMTRAJ-ARSCL's objective is to
       support studies focusing on those cloud decks while still providing analysis flexibility by running the trajectory calculations 5
days backward and forward in time. The ARSCL suffix in the dataset's name refers to the Active Remote Sensing of CLouds
       (Clothiaux et al., 2001), a widely-used ARM VAP, which combines data from ARM radars and lidars to produce an objective
       determination of cloud deck and hydrometeor vertical boundaries together with associated radar moments. In this context, a
       primary cloud deck can contain multiple liquid-bearing layers vertically connected by precipitation detectable by the profiling
       radar (e.g., Figure 1).
The ARMTRAJ-ARSCL dataset is initialized every 3 hours, similar to ARMTRAJ-SFC. The cloud deck base for trajectory
       initialization is determined as the 1-hour mean (starting at the initialization time stamps) cloud base height ("cloud base best
       estimate" field in ARSCL). This ARSCL field is processed using a ceilometer-micropulse lidar combination with a general
       tendency towards the ceilometer data product, which was previously evaluated against high spectral resolution lidar data and
       found to have a small positive bias typically under 50 m (Silber et al., 2018). This small bias should be, in most cases,
insignificant, given that cloud deck geometrical depths are commonly significantly greater (e.g., Lu et al., 2021). The cloud
       deck top is set as the 1-hour mean first radar top (first radar echo with an overlying clear-sky range gate), the samples of which
       are included in the averaging only if, at a given time step, they are above the cloud deck base. Among other similarities to

ARMTRAJ-PBL, we also run the trajectory calculations for the free troposphere to address community interest in processes such as cloud-top entrainment. Because the cloud top can be fairly variable over a 1-hour period, we set the free-tropospheric

height as the sum of the 1-hour mean cloud top, its 1-hour standard deviation (using the same samples as in the first radar-top averaging), and 200 m. Figure 1 exemplifies sets of cloud deck base, top, and free-tropospheric heights used to initialize the ARMTRAJ-ARSCL trajectories over a 24-hour period. Because a cloud deck was observed throughout the depicted day, the 3-hour initialization interval translates to the eight illustrated sets. Note the consistency between the cloud base markers and the initialized cloud deck base height, as well as the variable distance between the cloud deck top and free-tropospheric height,

depending on the cloud deck top's temporal variability.

Similar to ARMTRAJ-PBL, ARMTRAJ-ARSCL reports trajectory data for the cloud deck base, middle, and top and includes 99-member ensemble results using 11 equally distant heights between the cloud deck base and top combined with the same 3×3 grid as in the other datasets (9-member ensemble results are reported for the free troposphere). Hourly means of auxiliary data used and reported by ARSCL, such as hydrometeor field boundaries and liquid water path retrieved by ARMs microwave

radiometer (Morris, 2006) are included in ARMTRAJ-ARSCL data files.

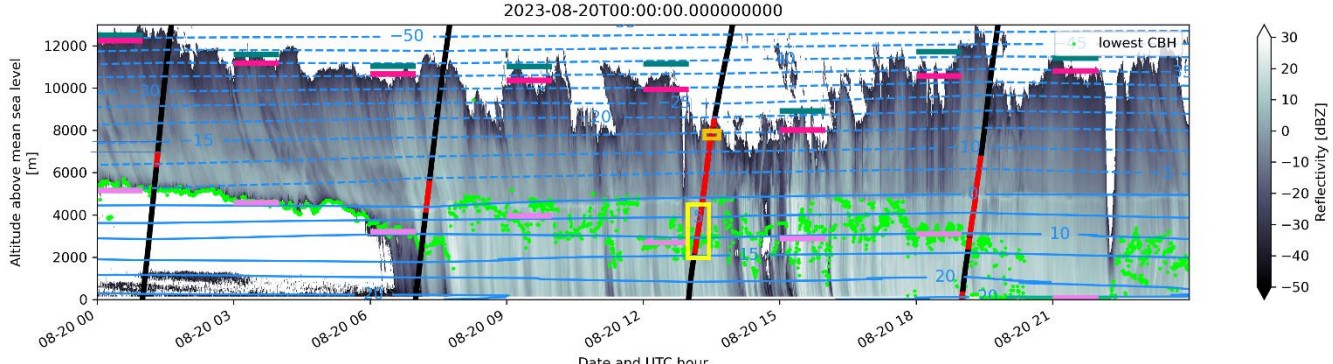

**Figure 1: ARSCL radar reflectivity (color-scale) on August 20, 2023, depicting the landfall of tropical storm Hilary at the ARM EPCAPE deployment in La Jolla, San Diego, California. The green markers denote the ARSCL-reported cloud deck base. Blue**
**contours represent temperatures (in degrees Celsius) from the ARM interpolated-sounding VAP (Fairless et al., 2021). Slanted black lines illustrate sounding profiles, with the red sections delineating sounding-based liquid-bearing cloud layer detections. The violet, pink, and teal horizontal lines designate the cloud deck base, top, and free-tropospheric heights for ARMTRAJ-ARSCL's HYSPLIT trajectory initialization. The line lengths (a fixed one hour) represent the ARSCL data averaging period. The yellow and orange rectangles highlight the liquid-bearing cloud layers analyzed in sect. 3.1.**


## 3 ARMTRAJ Application Examples Using the EPCAPE Datasets

The ARMTRAJ datasets currently cover the full EPCAPE deployment, from February 2023 to February 2024 (ARMTRAJ-ARSCL starting from March 2023). Given the diverse potential usage options of ARMTRAJ, here we limit ourselves to four short analyses utilizing each of ARMTRAJ's datasets. We first describe a case study using ARMTRAJ-CLD where we

exemplify the value of ARMTRAJ's ensemble runs in evaluating trajectory confidence and uncertainties. We then briefly
present three bulk analyses of ARMTRAJ-SFC, ARMTRAJ-PBL, and ARMTRAJ-ARSCL.

**3.1 Case Study: Mid- and Upper-Level Liquid-Bearing Cloud Layers in Tropical Storm Hilary**

Hurricane Hilary was the first tropical cyclone to hit Southern California as a tropical storm since 1939. By chance, this landfall
occurred during the 1-year long ARM EPCAPE deployment, most instruments of which were operating during the event.
Profiling radar observations, for example, captured the cloud deck evolution over La Jolla from a cirrus-topped mixed-phase
cloud to a heavily-precipitating deep cloud deck with multiple embedded liquid-bearing layers, indicated by the sounding
measurements (Figure 1). Figure 2 depicts a 5-day back trajectory of a cloudy airmass detected using the 13 UTC radiosonde,
which extended from an altitude of ~2,300 to 4,250 m (yellow rectangle in Figure 1). The trajectory for the cloud middle
section starting at the ARM deployment coordinates appears largely collocated with the ensemble mean up to 4 days backward
in time (Figure 2, left). This trajectory consistency and relatively small variability in airmass ensemble coordinates are
indicative of the single site coordinates' trajectory being representative in this specific scenario.

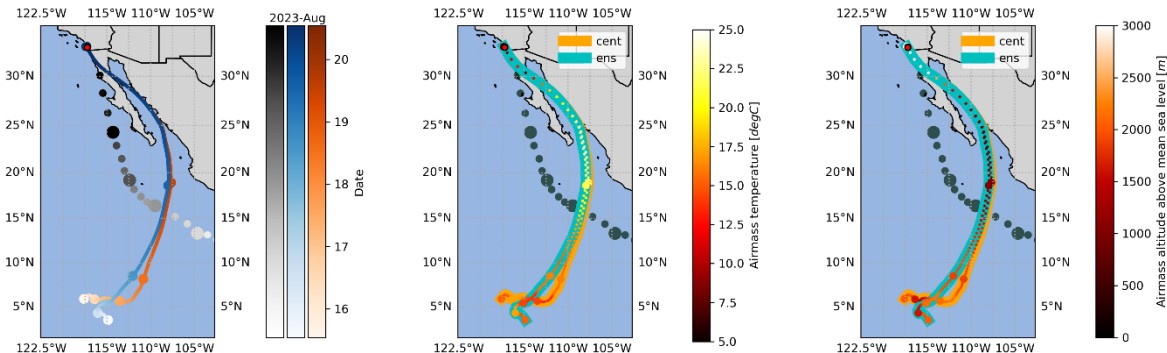

**Figure 2: ARMTRAJ-CLD 5-day back trajectory properties of a cloudy layer detected on August 20, 2023, using the 13 UTC**
**sounding measurements (see yellow rectangle in Figure 1). (Left) Airmass trajectory using the ARM deployment coordinates (orange**
**tints), ensemble-mean trajectory (blue shades), and 6-hourly markers of Hurricane Hillary's track (transitioned to a tropical storm**
**on August 20; grey tints). (Middle) The same trajectories overlaid with hourly airmass temperature and (right) altitude AMSL.**
**Larger markers denote 24-hour increments from the trajectory initialization time. The red marker designates the ARM EPCAPE**
**deployment site.**


Examination of the trajectory timing against the center of Hilary (Figure 2) suggests that the airmass entrained into the rear-
right flank of the cyclone roughly 1-2 days prior to the EPCAPE overpass. Forced by the cyclone, the airmass strongly
accelerated (increasing distance between large markers in Figure 2), and gradually subsided and warmed until ~18 hours from
the EPCAPE overpass (Figure 2, middle and right). From that point, being much closer to the cyclone center, the airmass was

lofted (Figure 2, right), cooled, likely adiabatically (Figure 2, middle), and eventually reached water vapor equilibrium, resulting in condensation.

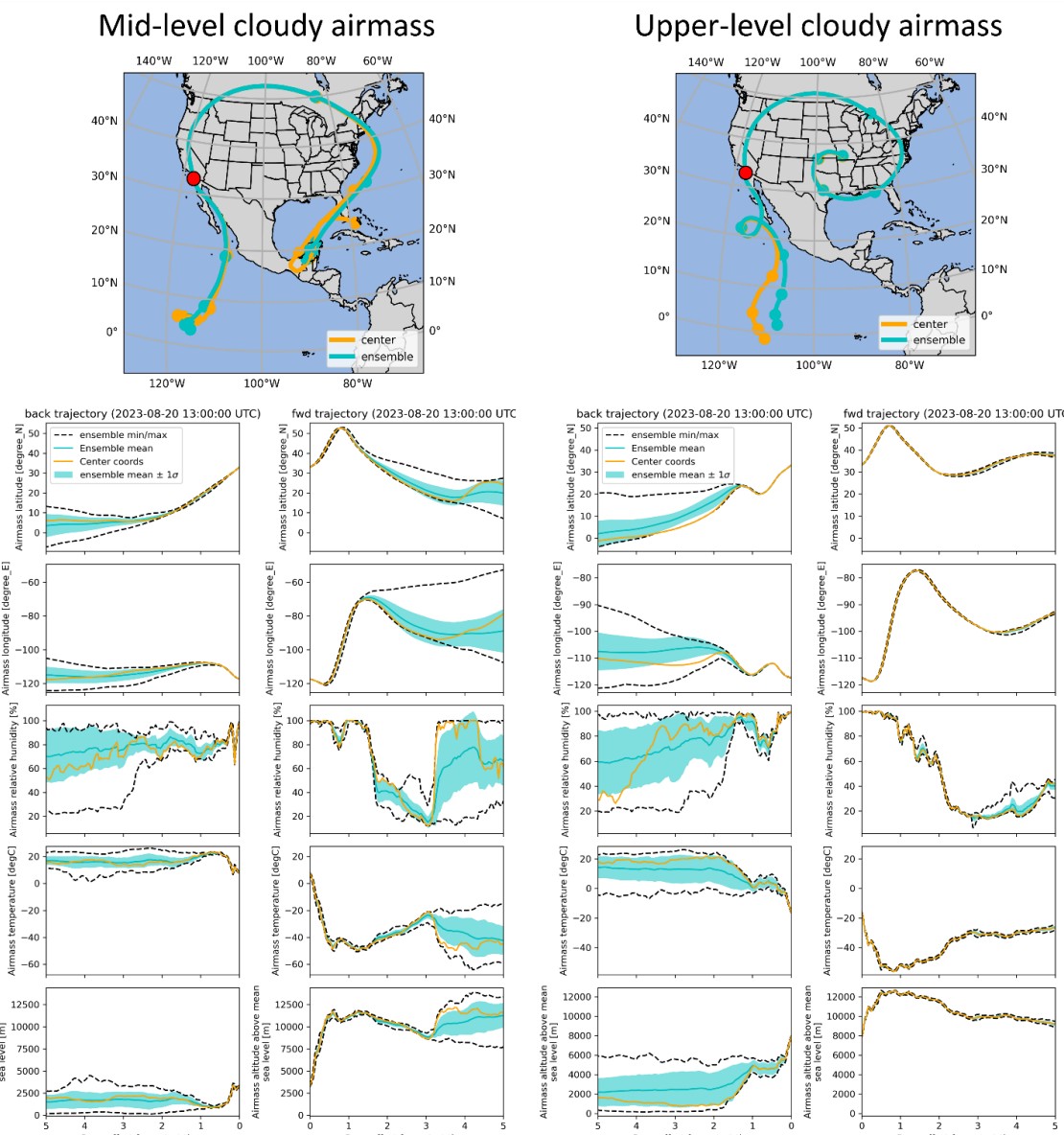

**Figure 3: Two sets of back and forward trajectories of cloudy airmasses detected on August 20, 2023 using the 13 UTC sounding.**
**The top two panels depict trajectory maps (larger markers denote 24-hour increments from initialization time), and the bottom panels illustrate (from top to bottom) time series of airmass latitude and longitude coordinates, relative humidity, temperature, and altitude AMSL. The back trajectory of the airmass in the left set of panels is the same as in Figure 2, whereas the right set of panels represents a cloudy airmass detected over the EPCAPE deployment at a higher altitude (see orange rectangle in Figure 1). Each time series plot shows the temporal evolution of airmass parameters along trajectories initialized at the center coordinates together**
**with the ensemble mean, minimum, maximum, and the mean ±1 standard deviation ($\sigma$) (see legend).**

The left set of panels in Figure 3 expands the analysis of center coordinates trajectories versus ensemble results by depicting time series plots of both back and forward trajectories of the same mid-level cloudy airmass discussed above. The right set of panels illustrates back and forward trajectory calculations initialized for a thin (~100 m deep) high-level supercooled cloud layer detected at ~7800 m using the same 13 UTC EPCAPE sounding profile (orange rectangle in Figure 1). In both cases, the airmasses are forced upward by the cyclone on the first day following the EPCAPE overpass, thereby cooling, producing more condensate, and converting into a cirrus cloud, as suggested from the airmass relative humidity, temperature, and altitude time series panels. Specifically, all ARMTRAJ ensemble members are consistently characterized by airmass relative humidity remaining at ~100% and temperature decreasing and staying below -38 °C. During this significant ascent of the cloudy airmasses to the tropopause region, they are entrained into the polar jet stream, which carries them several thousand kilometers (mostly eastward) in the following few days (see maps and the airmass latitude and longitude panels).

In the case of the back-trajectories for the mid-level cloudy airmass (left set of Figure 3 panels), the airmass parameters for the center coordinates and ensemble mean stay collocated for roughly four days, as noted above. However, the uncertainty of the airmass origin and thermodynamic properties generally increases backward in time, evident by the increasing ensemble standard deviation and the range between the ensemble member minimum and maximum. Similarly, the uncertainty of the forward trajectory parameters significantly increases starting roughly one day following the EPCAPE overpass. At the 5-day mark, the coordinate uncertainties are on the order of 10 degrees in latitude and longitude, and the ensemble member range is on the order of several tens of degrees; relative humidity uncertainty is ~20% and temperature uncertainty is greater than 10 °C, compared to ~5% and ~2 °C at the 3-day mark, respectively. Taken together, these ensemble results suggest low confidence in the airmass forward trajectory properties, especially beyond 2-3 days, and somewhat higher confidence in the airmass back trajectory properties.

The back trajectory ensemble spread in the right set of panels in Figure 3, representing the upper-level cloudy airmass is more extensive than the spread of the mid-level cloud layer discussed above. For example, 3 days prior to the EPCAPE overpass, this upper-level cloudy airmass exhibits relative humidity, temperature, and altitude uncertainties roughly double those of the mid-level cloudy airmass, with values of ~27%, ~10 °C and ~1830 m compared to ~14%, 4 °C, and ~850 m, respectively. However, given the ensemble temperature, relative humidity, and altitude largely monotonic tendencies, we can still deduce that this high-altitude airmass is most likely of warm and moist low-latitude low-level oceanic origin forced upward by the cyclone, as also suggested by the spiraling movement depicted in the top-right panel. Unlike the mid-level layer back trajectories (left set of panels), in this case, the center coordinates' airmass trajectory is one of the ensemble extremes at certain times; that is, even though the center coordinates are at the center of the ensemble latitude-longitude-height initialization mesh. As an additional contrast to the mid-level cloud case, the forward trajectory ensemble remains consistent with very little variance throughout the 5-day period. The differences in ensemble spread between those somewhat similar trajectories calculated for cloudy airmasses exemplify the case-specific nature of trajectory robustness and the value of ensemble data.

### 3.2 Bulk Analysis of Surface Back Trajectories

While the cloudy airmass observed during Hilary's EPCAPE overpass originated south to south-east of the deployment site, closer to the surface, the La Jolla region often experiences marine flow from the north-west directions (e.g., Liu et al., 2011). Here, we briefly examine the potential source origin and properties of airmasses reaching the EPCAPE deployment. We focus on ensemble mean variables, which are more robust than single trajectories for the deployment site coordinates, and provide uncertainty estimates, though these are largely excluded from this analysis for brevity.

Analysis of hourly-mean winds, taken in three-hour increments per ARMTRAJ-SFC's structure (sect. 2.1), indicates a westerly to north-westerly component dominance (Figure 4a). A joint probability density function (PDF) of 12-36 hour back-trajectory coordinates supports the surface measurement indications of marine airmass sources, specifically of coastal origin (Figure 4c). This proximity of airmasses to densely-populated regions could suggest that aerosol properties might be strongly influenced by the proximity to these more polluted regions, especially considering that more than 90% of hourly airmass samples in this

12-36 hour period were within the PBL (when accounting for ensemble standard deviation of airmass height).

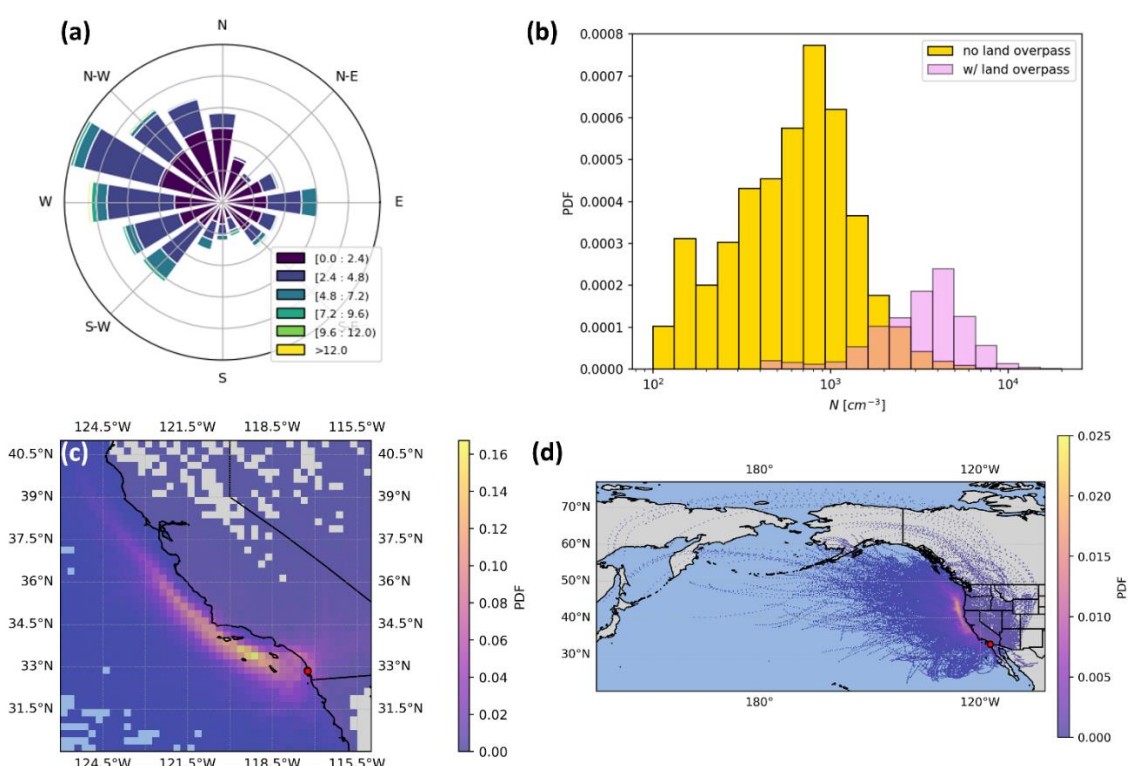

**Figure 4: ARMTRAJ-SFC bulk analysis: (a) Wind rose based on hourly-mean data from the ARM Surface Meteorology System (legend values are in m/s), (b) sub-micron total number concentration histogram (logarithmic bin width of 0.121) using SMPS data**
**partitioned based on whether 12-36 hour surface back trajectories had any land overpass (see text), (c) probability density function of 12-36 hour back-trajectory samples (bin dimensions of 0.25×0.25 degrees), and (d) same, but for 48-96 hour back-trajectories.**

Indeed, the distribution of sub-micron aerosols (10-500 nm in diameter) measured by the ARM scanning mobility particle sizer (SMPS; Kuang, 2016) illustrates a distinct picture. When partitioned based on whether a given 12-36 hour back trajectory

overpassed a land surface grid cell (defined here as an overpass of ERA5 grid cell with land area fraction exceeding 0.5 for at least 1 hour), a clear separation is observed in total aerosol number concentration with cases of airmasses overpassing land having notably higher concentrations (Figure 4b). This number concentration separation was also distinct when only trajectories corresponding to westerly surface wind direction measurements were used and, to a lesser extent, when the partitioning was performed using earlier periods such as 72-96 hours, 96-120 hours, etc. (not shown). This sensitivity to land

proximity, even in more distant periods in airmass hysteresis, is supported by the general consistency of surface airmasses sampled at EPCAPE to follow coastal flow patterns even several days before the arrival at the deployment site (Figure 4d).

### 3.3 Potential PBL Airmass Aerosol Sources Based on Bulk Statistics

We can further evaluate potential aerosol sources by examining surface and vegetation types airmasses overpassed. Here, we consider that aerosols are continuously mixed within the PBL along their trajectory path until they are eventually sampled at

the ground-based deployment site. We use the ARMTRAJ-PBL ensemble mean path and generate a pie chart of average surface and vegetation properties (implemented in the IFS model and reported in ARMTRAJ) along the trajectories during the 5 days preceding the arrival at EPCAPE's coordinates. We only count samples where the airmass ensemble mean height minus the ensemble mean standard deviation is within the PBL (~92% of all ARMTRAJ-PBL samples). Vegetation and surface property samples are weighted based on their corresponding cover fraction. The resulting pie chart (Figure 5) indicates the

dominating marine sources. This result is generally expected given the marine source dominance suggested in the ARMTRAJ-SFC combined with the median (average) PBLH of 180 (255) m calculated from the ARM surface measurements. On average, airmasses are influenced more than 4% of the time by Evergreen shrubs and needleleaf trees, known to be a significant sources of natural volatile organic compounds that can form secondary organic aerosols (SOA) (see Guenther et al., 1995; Shrivastava et al., 2017). A detailed analysis beyond the scope of this study might be able to robustly characterize the influence of these

airmass overpasses on surface aerosol observations.

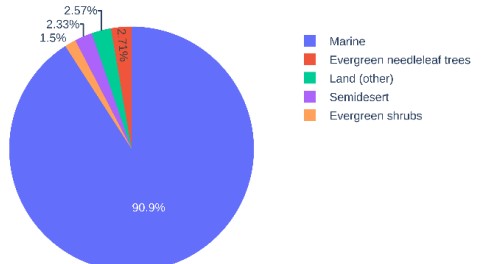

**Figure 5: ARMTRAJ-PBL bulk analysis of average surface and vegetation properties overpassed by EPCAPE PBL airmasses (see text).**

### 3.4 Bulk Analysis of Potential Airmass Origin Differences Between Cloud-Deck and Free-Tropospheric Airmasses

Aerosols such as SOA are commonly formed and transported within the PBL or other elevated mixed layers, serve as cloud condensation nuclei, and eventually influence cloud properties. Due to different hysteresis (flow patterns, atmospheric residence time due to chemical reactions, scavenging, etc.), free tropospheric aerosols often differ in their properties and source origin from mixed-layer aerosols in the same atmospheric column. This source origin difference can be demonstrated by analyzing the difference in potential source origin between cloudy and free-tropospheric airmass trajectories in the ARMTRAJ-ARSCL dataset. Figure 6 qualitatively illustrates this in-cloud and above-cloud airmass trajectory difference using 3-5 day ensemble mean coordinate PDFs. Some specific trajectories can be observed in the plot as distinct patterns. Similar to ARMTRAJ-SFC, in-cloud airmasses tend to concentrate along the Pacific coast, though with greater spread. This spread is likely the result of variable flow patterns between cloud-deck boundaries, which only partially overlap with the near-surface flow patterns or the PBL depth in general, as defined by the bulk Richardson number method. Unlike the in-cloud deck airmasses, the above-cloud (or free-tropospheric) airmasses tend to originate (within the 3-5 day range) in northeastern Pacific marine areas generally west to southwest of the deployment site as well as inland regions along the Great Basin and Mojave Deserts and the California part of the Sierra Nevada range. We can expect that aerosol properties of these in- and above-cloud deck airmasses would be different. However, such detailed studies likely involving airborne measurements such as those collected during the Southern California Interactions of Low cloud and Land Aerosol (SCILLA) experiment complementing EPCAPE, are beyond the scope of this study.

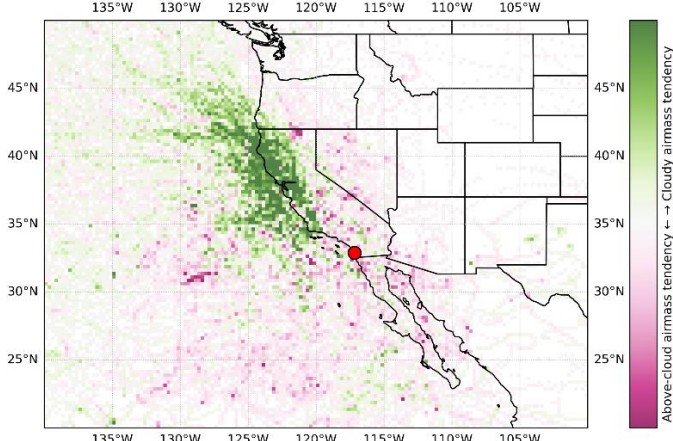

**Figure 6: ARMTRAJ-ARSCL cloudy (in-cloud deck) and free-tropospheric (above cloud deck) airmass source origin tendency qualitatively demonstrated by the difference between 3-5 day ensemble mean trajectory PDFs. The red marker denotes the EPCAPE main deployment site.**

## 4 Conclusions and Outlook

ARMTRAJ datasets provide essential support for the utilization of ARM deployments. They mitigate the gap ensuing from the typically fixed nature of ground-based deployments, give context to collected measurements, enable better synthesis of ARM observations with satellite observations, and could provide boundary conditions for modeling studies constrained by ARM measurements. Here, we showcased only a limited number of analyses that can be performed by synergizing ARM measurements with ARMTRAJ datasets. The case study example emphasizes the value of ensemble statistics provided in

ARMTRAJ datasets to evaluate uncertainties and level of confidence in the trajectory model results. It demonstrates that the level of confidence in trajectory calculations is case-specific but typically tends to decrease with the trajectory period and that conclusions drawn from a trajectory initialized at a single point can be misleading. We suggest that ensemble results should be preferred in most cases, especially when analyzing trajectories over several day periods.

While we presented an analysis of ARMTRAJ datasets generated only for the recently completed EPCAPE field campaign,

the implementation and application of ARMTRAJ will not end at that single site. ARMTRAJ recently reached an operational level such that datasets will begin production for past, ongoing, and future ARM deployments through the ARM infrastructure and will be continuously updated and made available via the ARM Data Discovery, with near-real-time production of fully annotated files.

## Data Availability

Current and future releases of ARMTRAJ datasets (Silber, 2024a, b, c, d) are and will be available on the ARM Data Discovery (https://adc.arm.gov/discovery/#/results/s::armtraj). Sounding (Keeler et al., 2022), meteorological station (Kyrouac et al., 2021), PBLH (Zhang, 2021), interpolated sounding (Jensen et al., 2021), ARSCL (Johnson et al., 2014), and SMPS (Kuang et al., 2021) data from the ARM EPCAPE deployment are available on the ARM Data Discovery (https://adc.arm.gov/discovery/; last access: 14 April 2024). Hurricane Hilary's tracking data (Kruk et al., 2010) are available at the International Best Track

Archive for Climate Stewardship (IBTrACS) website (https://doi.org/10.25921/82ty-9e16).

## Author Contribution

Conceptualization, formal analysis, investigation, methodology, visualization, and original draft preparation: IS
Project administration: JMC
Data curation and validation: IS and MRK

Manuscript review and editing: IS, JMC, MRK, and LMR

**Competing Interests**

The authors declare that they have no competing interests

**Acknowledgments**

The authors thank Krista Gaustad, Damao Zhang, John Shilling, Peng Wu, Jingjing Tian, and Scott Giangrande for valuable feedback. This research was supported by and data were obtained from the ARM user facility, a U.S. Department of Energy (DOE) Office of Science user facility managed by the Biological and Environmental Research (BER) program. The authors gratefully acknowledge the NOAA Air Resources Laboratory (ARL) for the provision of the HYSPLIT transport model used in this publication.

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
