# Peer review of "ARMTRAJ: A Set of Multi-Purpose Trajectory Datasets Augmenting the Atmospheric Radiation Measurement (ARM) User Facility Measurements"

_Earth System Science Data, 2024_

## Referee Comment (RC2)

In this manuscript, the authors present a set of datasets providing back- and forward airmass trajectories linked to ARM ground site measurements. The increasing use of Lagrangian analyses for studies of aerosols, clouds and other atmospheric phenomena is highlighted as a growing area of research interest, but performing these studies requires knowledge of trajectory models and the data to run them. Providing pre-processed trajectory data connected to ARM products will greatly help expand the ability of researchers to perform these analyses, as well as help improve the reproducibility of these studies. Overall, the manuscript is clear and well written, and the datasets are easy to access.

The authors present a number of case studies which are clear and provide good examples of the intended use cases for each of the separate trajectory datasets. However, while the issues of uncertainty in Lagrangian trajectories are mentioned in the manuscript they are not discussed in detail. With the long time-span of some trajectories, I would expect that the uncertainty becomes large in many cases and so care needs to be taken. As these datasets are intended to be used by researchers who may not have personal expertise in Lagrangian trajectory modelling, I think it would be particularly important to include a discussion on uncertainty and under what conditions the trajectories are expected to be more or less reliable. An additional section explaining these issues, possibly along with some supplementary figures, would greatly enhance the manuscript.

Specific comments:

Table 1: The "Initialized at" column for the ARSCL trajectories is a little difficult to parse, I suggest changing to "11 equally distant heights between the hourly mean cloud base and top for the lowest (typically primary) cloud layer".

Line 65: How does the vertical resolution of the ERA5 pressure level data affect the accuracy of the trajectories? Is it sufficient for more unstable conditions, particularly within the PBL? I am aware however that ERA5 model level data can't fit within the ARL files used by HYSPLIT, so it might not be possible to resolve this issue.

Line 110: This mentions tests performed to evaluate the uncertainty of longer back trajectories, but are not shown. It would be very nice to have these tests included as supplementary materials.

Line 219: For clarity: "we limit ourselves to exemplify 4 short analyses" -> we limit ourselves to four short examples

Line 229: Correction: "collimated" -> collocated

Figure 2: The shading along the trajectories in the middle and right figures is difficult to see. It may be clearer to present these as simple time series plots with the leftmost panel showing the spatial extent of the trajectory. More descriptive colorbar labels (e.g. "Hourly mean air temperature [°C]".

---

## Author Response (AR1)

**Author Responses**

We thank the reviewers for their valuable comments and helpful suggestions, which helped us improve the quality of the manuscript. To address reviewer comments, we reprocessed all four datasets after implementing a newer HYSPLIT version (v5.3 instead of v5.2) and ERA5 surface roughness data in trajectory calculations. Figures 2 onward were regenerated using the reprocessed datasets, but the differences from the previous submission are rather minor both qualitatively and quantitatively. Our responses and revisions are enumerated below. Beyond changes in response to reviewer comments, we have also made minor copy-edits.

**Reviewer #1 (Comments to Author):**

The primary concern with the manuscript is the inadequate assessment of uncertainties and the lack of a comprehensive quality check of the datasets. Although ensemble runs are run to indicate uncertainty, this primarily addresses uncertainties associated with the coarse resolution of meteorological fields. I recommend that the authors elaborate on uncertainties in the ERA5 data, uncertainties in the ARM measurements used to initiate the HYSPLIT, and propagation of uncertainties through the modeling processes. Additionally, it is not clear for example, whether the trajectory products on land cover and thermodynamic variables exhibit similar uncertainty levels, and how these uncertainties vary with the length of the backward or forward trajectory period (e.g., 2 days vs. 10 days). Addressing these aspects is crucial for users applying these datasets and interpreting results.

We understand the reviewer's concern and agree that a discussion about trajectory uncertainties was missing from the text. Below is an elaboration on each of the uncertainty aspects the reviewer raised:

1. ERA5 – this reanalysis data product is ostensibly considered "the gold standard" of reanalysis data products. There are already hundreds, if not thousands, of ERA5 evaluation studies (as a reference, the ERA5 article has more than 15,000 citations to date), a few of which were led by the lead author of this manuscript. Now, while uncertainties and errors necessarily exist, given the parameterized nature of the ECMWF's IFS model driving the ERA5 engine and the incomplete observational data assimilated into the analysis, evaluating the uncertainty directly from this product is certainly not a task within the scope of this manuscript.

2. ARM measurement uncertainties for ARMTRAJ initialization:
   - ARMTRAJ-SFC: this dataset does not use ARM measurements for initialization but only as supplemental data (from the MET instrument).
   - ARMTRAJ-PBL: Uncertainty could mostly stem from the Richardson number threshold of 0.25 used for PBLH determination or different PBLH determination methods rather than the SONDE instrument uncertainties, which are quite low (see Holdridge, 2020). We already refer to PBLH determination methods in the manuscript and justification for using the 0.25 Richardson method:

     *"There are other methods to determine the PBLH, the radiosonde-based retrievals of which are reported in ARM's VAP (see Sivaraman et al., 2013) and included in ARMTRAJ-PBL. However, the utilized bulk Richardson number method and threshold value were evaluated by Seidel et al. (2012), who suggested they are suitable for both convective and stable PBLs, though we note that Zhang et al. (2022) recently suggested this method better compares to ceilometer-based PBLH determination method under*

*stable PBL conditions. Moreover, the same method and threshold values are consistent with the PBLH implementation in ERA5 diagnostics used here."*

- ARMTRAJ-CLD: We use an RH method that considers instrument uncertainty and was previously validated against accurate liquid cloud layer determination by lidars. For clarification, we reworded the sentences mentioning that:

  *"Set radiosonde samples as "cloud" if RH values exceed 96%. This threshold value considers the radiosonde vendor's uncertainty (Holdridge, 2020) and was previously validated using lidar-based cloud layer detections (e.g., Silber and Shupe, 2022; Silber et al., 2020, Fig. S1; Stanford et al., 2023, Appendix D). We have also qualitatively examined detection consistency using higher RH thresholds against other remote-sensing measurements for different observed cases (not shown) and came to the same conclusion regarding the 96% threshold value validity."*

- ARMTRAJ-ARSCL: We use the ARSCL cloud boundaries. The cloud-top determination is based on vertically-pointing Ka-band radar echoes, which is ostensibly the most common (excluding satellites) and reliable method (including satellites) for cloud-top detection. The uncertainty in this case can be largely treated as the radar range gate separation of 30 m. Cloud base is determined mostly using ceilometer data. Ceilometer datasets were previously found to have a common bias that is typically smaller than 50 m, which should, in most cases, have, at most, a small impact, given the typical geometrical depth of clouds. We now refer to that bias in the methodology text:

  *"The cloud deck base for trajectory initialization is determined as the 1-hour mean ... "cloud base best estimate" field in ARSCL. This ARSCL field is processed using a ceilometer-micropulse lidar combination with a general tendency towards the ceilometer data product, which was previously evaluated against high spectral resolution lidar data and found to have a small positive bias typically under 50 m (Silber et al., 2018). This small bias should be, in most cases, insignificant, given that cloud deck geometrical depths are commonly significantly greater (e.g., Lu et al., 2021)."*

3. Error propagation: In the trajectory model calculations, error propagation is most likely to stem from the time stepping. We use HYSPLIT's dynamic time stepping ranging between 1 and 60 minutes, which is updated every hour based on the meteorological fields, grid spacing and a particle propagation limit (in grid units). We tested the latter in the past and found its default value to be robust in the vast majority of cases (not shown). We now refer to this factor in the text when discussing ARMTRAJ-SFC:

   *"While some studies examined back trajectories extending even 15 days, tests we performed using an ensemble approach (not shown) suggested that trajectory dispersion consistently becomes so substantial that the airmass information is no longer consistent and robust. This dispersion is most likely driven by the propagation of errors stemming from multiple factors, such as the integration time step and the limited vertical resolution of the ERA5 pressure-level data used by HYSPLIT, especially near the surface."*

4. Surface roughness (sea vs. land effects): That is a very good suggestion. In our previous submission, we ran ARMTRAJ processing using HYSPLIT's default fixed roughness length value of 0.2 m, which better matches some land surfaces than water surfaces. We now incorporate surface roughness length data from ERA5 in HYSPLIT calculations. All four ARMTRAJ datasets were reprocessed and figures were regenerated accordingly using our

recently implemented newer HYSPLIT version (5.3). That said, both qualitative and quantitative changes are rather minor, with the more significant impact being a few important bug fixes ARL has implemented in HYSPLIT v5.3 rather than surface roughness, which has some influence, but a relatively minor one even in the ARMTRAJ-SFC dataset. The influence is particularly small when ensembles are used. The plot below exemplifies one of the cases we found with the more significant roughness length influence. This case is from February 1, 2023 using ARMTRAJ-SFC ensemble mean coordinates. As noted above, a more significant effect is expected over the ocean, where the default roughness length is much greater than that of water.

[Figure]

However, in bulk processing, the effects often become negligible, even when the version change is considered. For example, the distribution of average surface and vegetation properties (Fig. 5) is essentially the same, with a quantitative difference of a few percent.

[Figure]

The second paragraph of section 2 was modified to reflect the incorporation of surface roughness information:

*"Trajectory calculations are performed with HYSPLIT, reading the same ERA5 global data files in pressure-level vertical coordinates supplemented with single-level reanalysis data fields such as PBLH and surface altitude, winds, and roughness length. Each ARMTRAJ dataset is initialized and configured differently to align with its purpose, potential use, and the characteristics of the ARM dataset required for initialization (see Table 1). ARMTRAJ datasets are discussed in detail below."*

We acknowledge that trajectories could be, in some cases, highly uncertain, and therefore, we decided in the first place to include the ensemble run statistics in all ARMTRAJ datasets. As mentioned above, we agree with the reviewer that a proper discussion was missing. To address this and reviewer #2's comments, we added a figure and a few paragraphs to section 3.1 discussing this topic, where we also relate to the reviewer's important note concerning the uncertainty as a function of trajectory period. The added figure and text are given below:

[revised manuscript text omitted]

Introduction**:** The manuscript adequately explains the utility of ARM observations and air mass trajectory analysis. However, given the common usage of the HYSPLIT model in prior studies, it would be beneficial for the authors to clarify what unique insights these new datasets provide, and what specific scientific questions they enable users to address beyond the capabilities of traditional HYSPLIT applications.

We wish to clarify that the new ARMTRAJ datasets leverage existing HYSPLIT capabilities (the trajectory model), so it is not clear to us what the reviewer refers to as "traditional HYSPLIT applications." To address the reviewer's comment based on our understanding, we added to the final paragraph of the Introduction some examples of scientific objectives that can be examined using ARMTRAJ:

*"Here we describe ARMTRAJ … which can be used to close some of the gaps ensuing from the Eulerian nature of many ARM cloud, aerosol, and other atmospheric measurements, thereby enhancing the versatility of ARM datasets. For example, understanding the impact of pollution upwind of ARM deployment sites on measured aerosol properties versus clean upwind conditions; the effect of aerosols entrained at cloud top on sink processes in clouds observed over ARM sites using ARM measurements as observational constraints on model simulations initialized using ARMTRAJ data; and the estimation of cloud lifecycles before and after overpassing ARM sites by synthesizing ARM, satellite, and ARMTRAJ data."*

2.1 Surface Trajectory Dataset: There is a potential issue with trajectories initiating at low altitudes as they may hit the ground and lose accuracy. Have the authors observed this issue in their datasets? A discussion on the impact of terrain on data quality would enhance this section.

That is a good point. Indeed, trajectories hitting the ground lose information about vertical motion, but they do not stop when intersecting the surface. They can still be transported horizontally and forced upward in subsequent times. This converges to our self-definition of what is an airmass for trajectory purposes. If it is merely a tracer, then indeed, we might want to initialize the trajectory model at some level above the surface, but this also does not guarantee that we will prevent such cases, and in many aspects, the ARMTRAJ-PBL dataset can serve that purpose. In any case, the ensemble runs mitigate such potential issues.

To further support our initialization choice, we performed a simple analysis using the full ARMTRAJ-SFC dataset for the center coordinates. The left panel below depicts the percentage of trajectory samples that hit the surface at each back trajectory time step. At 0 h (initialization), all samples are at the surface (100%), but at 24 hours, only 5% of samples "touch" the surface (see zoomed-in middle-panel plot), supporting the notion that such events are the exception rather than the rule. Even if we use the most permissive option of analyzing the height of the lowest ensemble

member at each time step (height_ens_min field in ARMTRAJ data), we still drop below 10% of samples relatively quickly.

This point that trajectories hitting the surface are the "exception rather than the rule" is manifested, together with other factors, in the fact that the mean height of trajectories consistently increases with time (backwards), and even the ensemble minimum height increases to a few hundred meters before stabilizing (see right panel).

[Figure]

We modified the text to reflect this discussion:

1. In the final Introduction paragraph where we mention the ensemble runs:
   *"Varying-size ensemble run results are also reported, facilitating the evaluation of trajectory consistency, robustness, and uncertainty, while mitigating potential near-surface artifacts and errors."*
2. When discussing the ARMTRAJ-SFC dataset – see quote concerning error propagation in our response to the first comment
3. In the description of the ARMTRAJ-PBL dataset (we shortened a sentence):
   *"The ensemble in the ARMTRAJ-PBL dataset is much greater than ARMTRAJ-SFC's ensemble… This extensive ensemble configuration ameliorates the lack of explicit mixing in the ECMWF Integrated Forecasting System (IFS) model used to drive ERA5 and, the limited near-surface resolution (~250 m) at the ERA5 pressure level grid."*

Table 1: Identification of potential applications for backward and forward ARMTRAJ-CLD and ARMTRAJ-ARSCL trajectories would provide clarity and usefulness to the readers.

Good idea. Thank you. We added a "potential application examples" column to Table 1.

**198: The mention of "the resulting 8 sets of cloud deck base, top, and free-tropospheric heights" is unclear. For better context, consider placing the sentence from paragraph #190 before this statement.**

We rewrote this sentence and now refer to the context of the sentence in l. 190 discussing the 3-hour intervals:

*"Figure 1 exemplifies sets of cloud deck base, top, and free-tropospheric heights used to initialize the ARMTRAJ-ARSCL trajectories over a 24-hour period. Because a cloud deck was observed throughout the depicted day, the 3-hour initialization interval translates to the eight illustrated sets."*

Figure 4, please include standard deviation of these numbers.

We understand the importance of providing uncertainties in different figures. That said, this figure is supposed to be simple, merely depicting an average surface and vegetation type for the entire EPCAPE dataset. Because the depicted variable is categorical and not quantitative, the standard

deviation is meaningless in this case. One could estimate uncertainties using various assumptions applied to individual trajectories or treating the category probabilities as Bernoulli distributions. However, we think such an analysis would render this demonstration and its related text ostensibly much more complex than they need to be, potentially missing the purpose of this simple example.

**Reviewer #2 (Comments to Author):**

The authors present a number of case studies which are clear and provide good examples of the intended use cases for each of the separate trajectory datasets. However, while the issues of uncertainty in Lagrangian trajectories are mentioned in the manuscript they are not discussed in detail. With the long time-span of some trajectories, I would expect that the uncertainty becomes large in many cases and so care needs to be taken. As these datasets are intended to be used by researchers who may not have personal expertise in Lagrangian trajectory modelling, I think it would be particularly important to include a discussion on uncertainty and under what conditions the trajectories are expected to be more or less reliable. An additional section explaining these issues, possibly along with some supplementary figures, would greatly enhance the manuscript.

That is a great suggestions. We agree with the reviewer that a discussion about trajectory uncertainties was missing from the text. This concern was also raised by reviewer #1. Determining under which conditions the trajectories are expected to be more or less reliable does not have a trivial solution and requires some dedicated research. That said, the reviewer is correct that in general, the longer the time span, the greater the uncertainty. Fortunately, the ensemble statistics provided by ARMTRAJ enable users to evaluate the reliability of specific trajectories or apply certain conditions in bulk analyses. We tried to incorporate and reflect all of these essential points in the new figure 3 and its accompanying discussion added to section 3.1. The new figure and text are given above in our response to reviewer #1's first comment.

Per the reviewer's comment, we now also provide general guidance to users in the Conclusions section:

*"Here, we showcased only a limited number of analyses that can be performed by synergizing ARM measurements with ARMTRAJ datasets. The case study example emphasizes the value of ensemble statistics provided in ARMTRAJ datasets to evaluate uncertainties and level of confidence in the trajectory model results. It demonstrates that the level of confidence in trajectory calculations is case-specific but typically tends to decrease with the trajectory period and that conclusions drawn from a trajectory initialized at a single point can be misleading. We suggest that ensemble results should be preferred in most cases, especially when analyzing trajectories over several day periods."*

Table 1: The "Initialized at" column for the ARSCL trajectories is a little difficult to parse, I suggest changing to "11 equally distant heights between the hourly mean cloud base and top for the lowest (typically primary) cloud layer".

Changed to a similar phrasing to that suggested by the reviewer: *"11 equally distant heights between the hourly mean base and top of the lowest (typically primary) cloud deck"*

Line 65: How does the vertical resolution of the ERA5 pressure level data affect the accuracy of the trajectories? Is it sufficient for more unstable conditions, particularly within the PBL? I am aware however that ERA5 model level data can't fit within the ARL files used by HYSPLIT, so it might not be possible to resolve this issue.

Indeed, we communicated about this pressure-level vs. model-level issue in the past with a few ARL personnel, but it appears that there is no solution at the moment or in the near future.

Line 110: This mentions tests performed to evaluate the uncertainty of longer back trajectories, but are not shown. It would be very nice to have these tests included as supplementary materials.

We agree. Given the new Figure 3 and its associated comprehensive discussion in response to the reviewer's first comment, however, we think that supplementary material would be redundant at this point.

Line 219: For clarity: "we limit ourselves to exemplify 4 short analyses" -> we limit ourselves to four short examples

Done. Thank you.

Line 229: Correction: "collimated" -> collocated

Corrected.

Figure 2: The shading along the trajectories in the middle and right figures is difficult to see. It may be clearer to present these as simple time series plots with the leftmost panel showing the spatial extent of the trajectory. More descriptive colorbar labels (e.g. "Hourly mean air temperature [°C]".

We updated the middle and right panels in Figure 2 to address the reviewer's comment:

1. Modified the used colormap to improve contrast.
2. Increased line thickness for better differentiation between the illustrated trajectories.
3. Modified the colorbar labels to be more descriptive.

The updated figure is provided below for reference:

[Figure]

The time series suggestion is excellent, and we implemented it in the new Figure 3, which presents both backward and forward trajectories together with their uncertainties.

---

## Author Response (AR2)

**Author Responses**

We thank the editor for the review and comments concerning the revised manuscript. Our responses and revisions are enumerated below. We would like to note that we made a minor text edit referring to ARMTRAJ's operational product. This edit reflects the progress that has been made in the implementation of ARMTRAJ in the ARM infrastructure since our initial manuscript submission. Thus, in the abstract, we changed "*ARMTRAJ is expected to become…*" to "*ARMTRAJ will soon become…*"

L65: the phrase "the effect of aerosols entrained at cloud top on sink processes in clouds observed over ARM sites using ARM measurements as observational constraints on model simulations initialized using ARMTRAJ data; " is hard to understand please rephrase.

Phrase changed to: "*evaluating the effect of cloud-top aerosol entrainment on sink processes by using ARMTRAJ data and ARM measurements to initialize and force model simulations; and analyzing …*"

L110: In my opinion, the following sentence requires a citation: "surface measurements are typically the most informative 110 about the sampled aerosol chemical, morphological, microphysical, and radiative properties"

We extended the sentence to clarify that with surface measurements, we do not face the same limitations such as payload properties as in the case of airborne systems: "*… surface measurements are typically the most informative about the sampled aerosol chemical, morphological, microphysical, and radiative properties given fewer limitations such as payload dimensions and weight.*"

We could not find references that explicitly state that. We could have added citations for papers describing airborne systems, but we suspect that readers might miss the direct connection to the text.

L116 consistently -> often (to avoid repetition of the word consistent)

Good point. Changed to "*predominantly*".

L116 and -> nor

Done.

L179 "Concatenate "cloud" samples distant by less than 50 m from each other. " are you referring to distance in the vertical, please clarify

Yes. Modified to "*vertically distant by …*"

L188 Many studies, e.g., marine stratocumulus cloud studies -> Many studies, such as those on marine stratocumulus clouds…

Done. Thanks.

L188 referred to here as … -> which in this context refer to the optically …

Done.

L202 greater -> larger

We think that "greater depths" would be a better fit in this context and therefore prefer to leave the text as is.

Figure 1: CBH is not defined, please do so

Unless we misunderstood the editor's intention, the figure caption already notes that the green markers represent the cloud base: "*The green markers denote the ARSCL-reported cloud deck base.*"

L231"ARMTRAJ,s 4 datasets" -> ARMTRAJ,s datasets. (too many "fours" in the sentence making it confusing)

Done.

Figure 2: please define the meaning of the bigger circles in the hurricane markers

Unless we misunderstood the editor's intention here, this is already noted in the figure caption: "*Larger markers denote 24-hour increments from the trajectory initialization time.*"

Please use center and ensemble as in Figure 3 for the legends in the middle and right panels.

We originally had legends in each panel, but this resulted in significant overlap with the plotted curves, and therefore we eventually decided to retain a single legend in each "panel set".

L260: is the sounding observation shown somewhere. I think it should be part of the Figure 2.

We removed from the text that part of the sentence mentioning the sounding. The sounding-based cloud layers were already discussed above and presented in Figure 1.

Figure 3: This figure is really confusing because the titles on the left and the right are the same. Please include the altitude launching height to differentiate the two

We agree with the editor and understand the concern. We added a large title to each panel set to accommodate this comment. The updated figure is shown below.

[Figure]

L272: "the same mid-level cloudy airmass" -> a mid-level cloudy airmass

Since that is the same cloudy airmass discussed above (Fig. 2) we now clarify that in the text: "*... of the same mid-level cloudy airmass discussed above*".

L289: How about the confidence in the back trajectory? Please add a similar sentence.

The sentence was extended to address this point:

*"Taken together, these ensemble results suggest low confidence in the airmass forward trajectory properties, especially beyond 2-3 days, and somewhat higher confidence in the airmass back trajectory properties."*

L337: "surface and vegetation properties (implemented in the IFS model and reported in ERA5)" are these reported in the ARMTRAJ data? If yes, please make it clear that this information is included, if not, please explain how to procure it, etc

Yes, this information is included in ARMTRAJ. We corrected the text: *"surface and vegetation properties (implemented in the IFS model and reported in ARMTRAJ) …"*